# Allosteric Binding of MDMA to the Human Serotonin Transporter (hSERT) via Ensemble Binding Space Analysis with ΔG Calculations, Induced Fit Docking and Monte Carlo Simulations

**DOI:** 10.3390/molecules27092977

**Published:** 2022-05-06

**Authors:** Ángel A. Islas, Thomas Scior

**Affiliations:** 1Vicerrectoría de Investigación y Estudios de Posgrado, Benemérita Universidad Autónoma de Puebla, Puebla 72592, Mexico; 2Laboratory of Computational Molecular Simulations, Departamento de Farmacia, Benemérita Universidad Autónoma de Puebla, Puebla 72592, Mexico; tscior@gmail.com

**Keywords:** ecstasy, entactogen, psychoactive, Molly, serotonin, benzofuran, NPS, cathinone, bath salts, cocaine, methamphetamine, designer drugs, escitalopram, antidepressant, 3D-RISM, electrostatic complementarity, XED, free energy, Gibbs free energy, citalopram, Cresset

## Abstract

Despite the recent promising results of MDMA (3,4-methylenedioxy-methamphetamine) as a psychotherapeutic agent and its history of misuse, little is known about its molecular mode of action. MDMA enhances monoaminergic neurotransmission in the brain and its valuable psychoactive effects are associated to a dual action on the 5-HT transporter (SERT). This drug inhibits the reuptake of 5-HT (serotonin) and reverses its flow, acting as a substrate for the SERT, which possesses a central binding site (S1) for antidepressants as well as an allosteric (S2) one. Previously, we characterized the spatial binding requirements for MDMA at S1. Here, we propose a structure-based mechanistic model of MDMA occupation and translocation across both binding sites, applying ensemble binding space analyses, electrostatic complementarity, and Monte Carlo energy perturbation theory. Computed results were correlated with experimental data (r = 0.93 and 0.86 for S1 and S2, respectively). Simulations on all hSERT available structures with Gibbs free energy estimations (ΔG) revealed a favourable and pervasive dual binding mode for MDMA at S2, i.e., adopting either a 5-HT or an escitalopram-like orientation. Intermediate ligand conformations were identified within the allosteric site and between the two sites, outlining an internalization pathway for MDMA. Among the strongest and more frequent interactions were salt bridges with Glu494 and Asp328, a H-bond with Thr497, a π-π with Phe556, and a cation-π with Arg104. Similitudes and differences with the allosteric binding of 5-HT and antidepressants suggest that MDMA may have a distinctive chemotype. Thus, our models may provide a framework for future virtual screening studies and pharmaceutical design and to develop hSERT allosteric compounds with a unique psychoactive MDMA-like profile.

## 1. Introduction

Currently, there is great interest and much debate about the clinical use of MDMA (3,4-methylenedioxy-methamphetamine). Double-blind phase 3 trials and meta-analysis of its use in assisted psychotherapy to treat post-traumatic stress disorder (PTSD) have yielded promising results. Hence, its approval by the Food and Drug Administration (FDA) appears to be imminent [1,2,3,4,5]. Despite this topicality, research on the actions of MDMA at a molecular level [6,7,8] remain rare.

The physiological and psychological effects of MDMA stem from its ability to cause the release of dopamine, 5-HT and norepinephrine to supraphysiological levels through interactions with their reuptake transporters DAT, NET and SERT [9,10]. The 5-HT transporter (SERT) is particularly involved in the behavioural response to MDMA in humans [11]. In fact, amphetamine derivatives that are more selective to the SERT than to the DAT are more likely to induce entactogenic MDMA-like effects with lower abuse liability [12], thus having pharmaceutical potential. Accordingly, the MDMA-induced release of 5-HT in the nucleus accumbens via the SERT is necessary and sufficient to explain its prosocial effect, but not its non-social drug reward in rodents [13]. Moreover, MDMA promotes fear extinction learning in a SERT-dependent manner [14], providing a mechanistic basis for its beneficial effects in treatment-resistant PTSD [10].

The hSERT constitutes the molecular target of the majority of antidepressants and it is comprised of twelve helical transmembrane segments with two known binding sites: a central (S1) and an allosteric (S2) site. They are located midway along the transporter and at the extracellular vestibule, respectively. The SERT is a neurotransmitter Na^+^ symporter that terminates serotonergic neurotransmission by coupling the transit of Na^+^ and K^+^ along their electrochemical gradient, to the internalization of 5-HT into the presynaptic cell. Antidepressants impede 5-HT reuptake through the SERT by stabilizing an outward open conformation [10,15].

Most drugs and test substances bind S1 with higher affinity with the exception of vilazodone and ligand Lu AF60097. It is noteworthy that the occupation of the ‘low-affinity’ allosteric site has a synergistic effect, deterring the dissociation of antidepressants and potentiating the binding for the central site [16,17,18,19]. On the one hand, MDMA occupies the central site of the hSERT acting as a substrate, as it is transported across the presynaptic serotonergic neuron. This process is energetically coupled to the drug-induced reverse flow of 5-HT, i.e., the SERT-assisted transport from the cytosol of the pre-synapse into the synaptic cleft. On the other hand, MDMA also inhibits the reuptake of 5-HT in a similar manner to a typical antidepressant. It is believed that it is by this dual profile that MDMA elicits its psychoactive effects that are clinically valuable, e.g., the therapeutic bonding and trauma reprocessing without emotional distress [3,10].

Previously, we proposed a mode of action for MDMA at S1, characterising the ‘ensemble binding space’ throughout the hSERT conformational cycle on trajectories from interpolated elucidated structures. In addition, we correlated the docking interaction energies of MDMA and a set of metabolites and analogues to their 5-HT uptake experimental activities (EC_50_s), providing a pharmacophore model [8].

Here, to shed light on the mechanism of action of this entactogen, we tested all current experimentally elucidated structures of the hSERT to demonstrate how favourably it occupies the allosteric site. The concomitant MDMA occupation of both hSERT binding sites was investigated by a combination of ensemble binding space analysis with enthalpic-entropic ΔG calculations, energy perturbations with the Monte Carlo method, extra-precision semi-automatic induced fit dockings, 3D-RISM solvation thermodynamics and electrostatic complementarity calculations. Our modelling approach was validated with experimental data by correlating the 5-HT reuptake inhibitory activities of MDMA and a group of related psychoactive agents, to their computed ΔG values upon their binding to the two binding sites. Of note, we made use of the ‘Molecular Field point Technology’ developed by Cresset™ under the XED force field, that provides a multipole electron distribution at a near quantum level [20].

This structure-based characterization of the energetically favourable features of the allosteric binding of MDMA provides: (i) a dynamic mechanistic hypothesis of the early molecular events in the transport of this drug along the hSERT; (ii) a guidance for the design of further experimental and computational studies of the MDMA/hSERT interplay; (iii) a theoretical framework for virtual screening and rational design of MDMA-like novel hSERT allosteric modulators.

## 2. Results and Discussion

### 2.1. MDMA Allosteric Occupation of the hSERT Overlaps the Binding Site of Escitalopram

High-performance induced fit docking of MDMA was carried out on S1 and S2 of the hSERT in an outward open conformation (PDB:5I73). At the allosteric site, this phenylethylamine adopts a conformation, perpendicular to the *z*-axis (aligned to the lumen of the transporter), docking between transmembrane segments TM1, TM6 and TM10 (Figure 1A). The simulation of the system with the reference interaction site model (3D-RISM) provided a quantitative thermodynamic description of the aqueous environment in which MDMA binds the extracellular vestibule of the hSERT.

The methylammonium moiety of MDMA simultaneously formed a salt bridge and a H-bond with the backbone and side chain of Glu494, in addition to a H-bond with a water molecule that favourably accommodates inside the binding pocket. While the benzodioxol group of MDMA forms a cation-π and a parallel π-π interaction with Arg104 and Phe335, respectively (insert in Figure 1A). Of note, these three residues constitute key elements for the allosteric binding of escitalopram (Figure 1B), as the substitution R104K decreases its affinity for this transporter [17]. Likewise, mutant E494Q significantly reduced the potency of this antidepressant but did not affect that of imipramine, which also binds S2 [18].

Glu494 is also fundamental for the binding of the high-affinity allosteric inhibitor Lu AF60097 and importantly, Phe335 is part of a motif that propagates the allosteric communication between S1 and S2 [18]. Thus, the occupation of S2 by MDMA may be coupled to its binding on the orthosteric site and in turn may play a role in its mechanism of action on the hSERT.

Next, to validate our modelling results we tested the hypotheses of whether the 5-HT block activities of MDMA and a set of related psychoactive hSERT inhibitors could correlate with the estimated Gibbs free energies (ΔG) on either their central or their allosteric binding. We selected five benzofurans, one cathinone, methamphetamine, MDMA, and cocaine (Figure 2) and carried out the simulations on an outward open hSERT structure.

Of note, our estimations of binding free energies go beyond classical docking scores by including entropic, enthalpic, polar ligand desolvation, and non-polar solvent contributions (see methods Section 3.2).

### 2.2. Experimental-Computational Quantitative Correlation of 5-HT Reuptake Inhibitors to S1 and S2 of the hSERT

The ΔGs of the MDMA congeneric hSERT blockers, methamphetamine, and cocaine were calculated upon their molecular association to the central and the allosteric binding sites of the hSERT. In agreement with the affinity data for allosteric antidepressants [17,18], all compounds showed higher binding strength to S1 than to S2. The in vitro inhibitory activities of tritiated 5-HT via the hSERT, heterologously expressed in cell assays [7] were linearly correlated to their theoretical free energy values. Of note, the correlations were possible despite the use of racemic mixtures in the experiments, while only the (R)-enantiomers were used for calculations.

In line with previous experimental findings on other MDMA analogues [6], the occupation of S1, rather than S2 best explained the inhibitory potencies of this set of ligands (Figure 3A,B). Of note, the computed binding modes of the selected compounds in S1 only have Glu98 in common with the binding of classical antidepressants and novel hSERT inhibitors [21], with the exception of escitalopram that may interfere with Tyr176 and Ser336 [22]. In particular, the orientation and modes coincide with the previously described binding mode for MDMA (on the paroxetine-induced conformation) [8] with the addition of Ala96 and S336 side chain contributions that likely reflect the escitalopram-induced conformational rearrangement. Our proposed MDMA binding pose and binding residues were highly conserved among all compounds, even for cocaine and methamphetamine at S1 (Figure 3C). In contrast, the binding modes for S2 were more heterogenous (Figure 3D).

Notably, the docking solution of cocaine correctly explains why the mutant Y176C significantly decreases its affinity [23]. In stark contrast, methamphetamine with the lowest 5-HT inhibitory potency docked between S2 and S1 (in light green in Figure 3D). Our interpretation is that this reflects the propensities of the drugs to be internalized through the transporter eliciting the efflux of 5-HT, e.g., MDMA and methamphetamine or to merely block the neurotransmitter reuptake, e.g., cocaine.

Thereafter, we focused on characterizing the allosteric binding of (R)-MDMA on the conformational landscape of the hSERT given by experimentally determined structures, by carrying out an ensemble binding space analysis. This approach mimics the dynamic ligand-protein cooperation, accounting for protein flexibility and ligand mobility within the active site by considering the best alternative binding modes on an ensemble of protein conformations [24].

### 2.3. Ensemble Binding Space Analysis of MDMA on the Allosteric Site of the hSERT: 5-HT and Escitalopram Analogous Binding Modes

The simulations were carried out first on the six recently elucidated cryo-EM structures of the hSERT, which characterize the allosteric site of 5-HT. Importantly, these structures are believed to recapitulate all fundamental states of the neurotransmitter transport cycle from holo open outward, holo occluded, over holo inward open toward apo inward open states. Holo structures are in complex with two molecules of 5-HT at S1 and at S2, [25].

Several low-energy ‘escitalopram-like’ binding modes similar to the previously proposed in Figure 1A were identified on the outward open and occluded holo states (PDBs:7LIA and 7MGW), i.e., the conformations in NaCl in which the substrate internalization cycle begins. (Figure 4A). In this binding solution, it is only Glu494 that binds both 5-HT and MDMA by their cationic amines and again, the thermodynamic calculations indicate that it is probable that this moiety interacts with water (insert in discontinuous lines, Figure 4A). Intriguingly, at this conformation, MDMA lies deeper inside the extracellular vestibule of the transporter, protruding its ring system towards the sodium (~7.6 Å apart) located in between transmembrane helixes TM1a and TM1b, which also binds Asp98, a key residue for MDMA binding to the central site [6,8].

The ensemble binding space of MDMA at the allosteric site was analysed and parameterized with the 80 most energetically favourable binding solutions. Figure 4B shows the number of solutions obtained at each state and the parameters based on the calculated Gibbs free energies. Statistical metrics (mean, standard deviation, mode, and range values) indicated little ΔG disparity and reflected a heterogeneous ensemble of solutions, implying relevancy to the MDMA allosteric mechanism of action. The sensitivity score encodes the capacity of the ligand to vary its free energy values by adjusting its own conformation [24]. Compared with the ensemble binding space reported for MDMA at the central site, the sensitivity score for S2 denotes a higher intramolecular flexibility, as the range score was double of the one previously obtained for S1 [8]. Meanwhile, the average rmsd at S2 reflected a considerable exploration of the binding space or ‘binding mode displacement’. It is noteworthy that the parameters obtained here originate from more precise ΔG terms (see methods Section 3.2) rather than classical enthalpic docking scores previously obtained for the central site [8].

Additionally, in the previous work, an equal number of poses per structure was obtained, here only the highest scoring poses were considered. As a consequence, it was clear to see that MDMA preferentially binds the occluded and the outward open states, suggesting that the affinity of the drug increases as the hSERT transits from the latter to the former, being sequestered similarly to the endogenous substrate. Furthermore, although no energetically favourable poses were found for the transporter in the occluded apo state (PDB:7LI7), some were found for the inward open apo states with Na^+^ and K^+^ cations, conditions under which MDMA could initiate the reverse translocation of 5-HT.

Remarkably, the top binding solution from this analysis, identified in the occluded state, coincides with the orientation and binding mode of 5-HT. All solutions were clustered and the pose from the most populated cluster (highest number of solutions) interacted with three of the six 5-HT binding residues in this state (Figure 4C). The reduced distribution of thermodynamically stable water molecules at S2 in the occluded state compared with the outward open state accounts for the diminished solvent accessibility. However, in the occluded state, the ammonium groups of 5-HT and MDMA can directly interact with Asp328 or via H-bond networks of waters as it is predicted by the 3D-RISM to be heavily hydrated. In contrast with Glu494, which preserves the salt bridge with these monoamines, observed in the outward open state.

In addition, both ligands bind Tyr495, albeit in a different manner i.e., 5-HT forms a H-bond with its backbone and MDMA an edge-to-face π-π with its side chain (Figure 4C).

Moreover, the global results show that MDMA can interact with all six 5-HT binding residues in the occluded state and specify the individual bonding incidences along the whole conformational ensemble (Figure 4D). Clearly, the aforementioned salt bridges with the two anionic residues at TM10 and TM6 are critical for MDMA allosteric binding throughout the hSERT transport cycle. The high prevalence of the amine-Glu494 bonding in the allosteric binding of MDMA to hSERT is explained by the often simultaneous interactions with the side chain and backbone atoms of this glutamate. Likewise, Thr497 may form two concomitant H-bonds, and finally alternating aromatic interactions, e.g., with Tyr495, Phe556 and Tyr579, may also play a role in the occupation of the allosteric site by MDMA. Of note, Phe556 is also involved in the recognition of escitalopram at the allosteric site [26].

Taken together these results indicate that both the ‘escitalopram-like’ and the ‘5-HT-like’ MDMA binding modes at S2 may contribute to the allosteric mode of action of this psychoactive compound. In agreement with the literature, multiple binding modes, such as these two, can be exploited for the drug design of new allosteric compounds of the hSERT [27].

In the pursuit of a compelling theoretical pathway for MDMA along the hSERT, we then wondered whether these two symmetrically opposed configurations of MDMA could alternate within the S2 binding pocket.

### 2.4. Intermediate Poses between the ‘5-HT-like’and the ‘Escitalopram-like’ Binding Modes of MDMA

Indeed, low-energy binding solutions in between the ‘5-HT-like’ and the ‘escitalopram-like’ orientations of MDMA were found in the ensemble binding space analysis. The polar cavity in which these intermediate binding modes lie (Figure 5A,B) suggest the ligand is able to flip from one orientation to the other in situ. Most probably during the transition process from the outward open to the tightly bound occluded conformation. Since ‘the most intermediate conformations’ detected (in magenta, Figure 5B), i.e., almost parallel to the substrate pathway (*z*-axis) are exclusively present in the occluded state. The binding residues involved in such proposed reorientation from the outward open to the occluded state (at transmembrane helixes TM1b, TM6 and TM10) are shown in Figure 5C. Of note, the allosteric residue Phe335 drastically rearranges its side chain along the transition from one state to the other (black arrow in Figure 5C).

The following three steps provided a more thorough and dynamic characterization of the early allosteric MDMA binding events on the hSERT: (i) selection of the most energetically favourable MDMA/hSERT complexes, in the ‘5-HT-like’ and the ‘escitalopram-like’ configuration from the ensemble space; (ii) high-accuracy induced fit docking of this ligand to the central site yielding double-bound models occupying both S1 and S2; (iii) subjecting these models to stochastic energy perturbations with the MC method. These steps were followed with the aim of evaluating the stability of the MDMA-hSERT interactions quantitatively and to capture the protein rearrangements induced by occupation of the two binding sites.

Four energetically favoured poses from the ensemble binding space were chosen for the simulations, two in which MDMA is in the ‘5-HT-like’ binding mode and two in which it is in the ‘escitalopram-like’ orientation. A ‘control simulation’ was run with MDMA only at the central site to distinguish the protein intramolecular changes induced by the occupation of S2.

### 2.5. Monte Carlo (MC) Simulations on Double-Bound MDMA/hSERT Models to Identify Allosteric Determinants

The preservation and stability of the MDMA-hSERT intermolecular and intra-residue interactions along the MC trajectories in the lowest ΔG outward open models were assessed. To this end, the interatomic donor-acceptor, π-π or cation-π distances were monitored as a quantitative marker of bonding contacts (Figure 6A–C and Figure 7B,C). Figure 6D shows the two resulting complexes after the stochastic energy jumps in the ‘5-HT-like’ configuration and Figure 7D shows the models in the ‘escitalopram-like’ orientation.

The most frequently contacted amino acid in the full MDMA allosteric ensemble space (Figure 4D) was Glu494 and the prevalence of its interactions with the cationic amine of MDMA was corroborated. Accordingly, in three of the four MC simulations, an intermolecular salt bridge was preserved or acquired (in red and orange, Figure 6A), while, simultaneously in two of them, a backbone H-bond with this residue was conserved (in light and dark blue, Figure 6A). Of note, these interactions distinctly enhanced the intracellular Glu494-Arg104 salt bridge, slightly pulling TM10 and TM1b together and briefly inducing an extracellular Glu494-Lys490 interaction. In contrast, in one case, the loss of the MDMA-Glu494 salt bridge precluded the formation of the Glu494-Arg104 bond and coincided with the optimization of the Glu494-Lys490 ionic interaction (Figure 7B,C). Importantly, the conformational coupling between Glu494-Arg104 and Glu494-Lys490 is crucial to the allosteric inhibition of escitalopram [26], as well as for the binding of the allosteric SSRI vilazodone [19].

Likewise, the strengthening and stabilization of the cation-π attraction between Arg104 and the benzodioxol of the drug (Figure 6B in blue) in the simulations with the escitalopram-like orientation, also optimizes the Glu494-Arg104 interaction compared with the control MC simulation, to the expense of losing the Glu494-Lys490 bond in one system, while it is preserved and optimized in the other with the aid of Glu493, that also binds Arg104 (Figure 7B,C).

Aromatic bonds also prevailed along MC trajectories in both ligand orientations, the most solvent exposed ‘5-HT-like’ pose of MDMA formed a stable π-π stacking with Phe556 or Tyr495 and a short-lived π-π interaction with Tyr579, three residues involved in the allosteric binding of 5-HT (Figure 6B,D) but not escitalopram. Subsequent energy minimization of one of the ‘5-HT-like’ systems allowed to retain these interactions while forming a cation-π tie between the methylammonium of MDMA and the phenyl ring of S2-S1 allosteric propagating residue Phe335 [18]. Of note, this residue is also contacted by MDMA in one of the escitalopram systems albeit via a π-π contact displaced stacking, while retaining the electrostatic interactions with Ar104 that engages one of the oxygens of MDMA (Figure 6B and Figure 7A).

The comparison of the MC-induced conformational readjustments with those occurring in the occluded state (Figure 5C) suggest that these interactions may not only be key to the allostery of this compound but may participate in the substrate-induced transition from the outward open to the occluded state [18].

Lastly, the stability of the aforementioned Thr497 and particularly Gln332 H-bonds with the amine of MDMA was verified in the ‘5-HT-like’ and ‘escitalopram-like’ configurations, respectively. It is noteworthy that in the latter case, the salt bridge with Asp328, previously observed in the occluded conformation, may exist simultaneously in the outward open state (Figure 6C and Figure 7A). Aforementioned residues are also involved in the allosteric modulation of the hSERT, as the high affinity ligand Lu AF60097 but not escitalopram also H-bonds with the side chain of Gln332, while Thr497 caps the binding pocket of the allosteric SSRI vilazodone [19].

In short, the MC trajectories show how the presence of MDMA at S2 in the two symmetrically opposed orientations affect the conformation of the hSERT at sites critically involved in the allosteric regulation of this transporter. In turn, these results suggest that the binding of MDMA to the allosteric site may synergistically affect that of the central site. In addition, we revealed that MDMA shares molecular allosteric features with 5-HT, antidepressants, and allosteric ligand Lu AF60097, reflecting the electrophysiological evidence of its role as a substrate and as an uptake inhibitor [10]. Together, the ensemble binding space dockings and MC simulations imply that MDMA may constitute a unique chemotype for structure-based drug design.

We next wondered how likely it was for MDMA to navigate from the allosteric to the central binding site, in view of the small tunnel there is between them in the open outward state [25] that, nevertheless, is proposed to be a gateway for the bulkier escitalopram to reach the orthosteric site [26] and since some putative allosteric ligands elongate between the two sites [28].

### 2.6. Ensemble Binding Space Analysis and Electrostatic Complementarity of the Pathway of MDMA from S2 to S1

Pursuing the idea of a path for MDMA between the allosteric and the central site, we first carried out a binding space analysis sampling both active sites simultaneously on a double-bound complex subjected to MC simulations. Low-energy intermediate binding poses were identified (Figure 8A in grey thin sticks). Figure 8B shows how often these poses involved residues from the central site (in asterisks) and from S2.

In addition to previously identified ‘allosteric residues’ Arg104, Gln332 and Phe335, the contribution of Tyr95 along the proposed S2–S1 pathway stands out. The high contact frequency of this residue is due to alternating or simultaneous cation-π contacts and backbone H-bonding with the ionic head group of MDMA. Moreover, Tyr95 may play a dual role, in the allostery of MDMA and in its transport as a substrate, since it is a binding residue for the occupation of vilazodone at S2 [19] and its displacement is necessary for the release of 5-HT into the cytoplasm [25]. This binding space analysis revealed that the amine of MDMA persistently interacts with Asp98 of the central site, which seems to be the driving force for the entry of MDMA into the central site.

To account for protein conformational freedom, we next carried out an ensemble binding space docking, probing a more extensive region on all 26 available hSERT structures. The preferential ‘high-affinity’ binding of this drug to S1 was corroborated and intermediate poses were detected (Figure 8C).

On the one hand, the MC results in one of the systems with the escitalopram-like orientation suggest MDMA could access the orthosteric site adopting this orientation without visiting the ‘5-HT-like’ binding pathway (arrow in Figure 7A). On the other hand, the occurrence of an extracellular energetically favoured binding pose found in the occluded state (insert of Figure 8C) and some found in complexes bound to sertraline (PDBs: 6AWQ and 6AWO) reinforces the notion of the reorientation of the ligand at the vestibule of S2 (Figure 5B and Figure 8C). It is noteworthy that in the outward conformation of the hSERT, S2 and S1 are fused into one cavity. However, both sites are separated in the occluded state by the Tyr176-Phe335 gate [29].

Fundamentally, it is clear from the electrostatic complementarity calculations of the models subjected to MC energy perturbations (Figure 8D) that: (i) MDMA in an escitalopram-like orientation can traverse the narrow cavity between the allosteric and the central site, without the energetic cost of changing its orientation, perhaps driven by the protein conformational rearrangements that ensure the closing of the extracellular vestibule in the occluded state before the sequestering of the substrate [25]. (ii) Two MDMA molecules may concurrently occupy the allosteric site, one with the ‘5-HT’ and the other with the ‘escitalopram-like’ orientations (Figure 8D, insert). This way, the passage of one molecule from S2 to S1 is possible, while another remains bound, stabilizing allosteric changes, e.g., Glu494-Arg104 and Glu494-Lys490 coupled salt bridges [26], and possibly facilitating substrate internalization.

Finally, to challenge our allosteric models, we screened a set of hSERT active compounds [30] (Figure 9A). All 5-HT releasers fitted the MDMA ‘5-HT-like’ conformation (Figure 9B) and the ‘escitalopram-like’ binding mode of S2, albeit with statistically different ΔG mean values (Table 1).

These results emphasise the dual allosteric binding mode of 5-HT releasers via the hSERT, suggesting the contact with 5-HT binding residues is particularly important for the reverse flow of 5-HT.

## 3. Methods

### 3.1. Induced Fit and Ensemble Docking

The ligands and the proteins were prepared in Flare4.0.2 (Cresset^®^, Litlington, Cambridgeshire, UK) [20]. Missing hydrogens were added and optimal ionization states of ligands and protein residues were assigned (pH = 7), using the thermodynamic sampling of amino acid residues (TSAR) method, a graph-theoretical approach from BioMolTech (Toronto, ON, Canada) [31]. The spatial positions of polar hydrogen atoms were optimized to maximize H-bond interactions and minimize steric strain. The side-chain orientation of His, Asn, and Gln residues for which X-ray analysis can return flipped orientations due to apparent symmetry were optimized. Gaps in the protein sequences of 1 or 2 residues if any, were filled. Residues with unresolved sidechains were detected and reconstructed and atoms from residues with incomplete backbone if any, were removed. Steric clashes were relieved by allowing small sidechain movements. Protein chains were capped if truncated, native ligands were extracted and template ligands for induced fit docking were selected manually. No native water molecules were around the active site (see Section 3.2). However, ions, when present, were included in the simulations. Active site was defined as the set of residues within 6 Å of the template ligand. The three-dimensional structures of ligands were generated in Flare 4.0.2 and energy-minimized before docking.

High-precision flexible-receptor/flexible-ligand docking was performed with LeadFinder™ (BioMolTech) in Flare 4.0.2 under the XED force field by Cresset™. This patented second-generation force field redefines the charge toward a multipole electron distribution akin to a quantum orbital description [20]. The ‘Very Accurate but Slow’ mode was used. This increases the accuracy and reliability of predictions by performing induced fit dockings by triplicate, making use of the most rigorous sampling and scoring genetic algorithm search combined with multilevel local optimization procedures and smart exploitation of the knowledge collected during a search run. To simulate the process of induced fit: automatic ‘on-the-fly’ and manual post docking local rearrangements of the active site were carried out, in conjunction with energy minimizations, thereby: (i) identifying the best pre-existing complementarity and (ii) simulating multiple tentative collisions with mutually induced conformational adjustments of the interacting species to achieve the most appropriate match.

To account for active site flexibility and to simulate the conformational selection binding mechanism, ensemble docking was carried out in Flare4.0.2. This enables the inclusion of multiple protein structures in the same docking run. Proteins were superimposed and the binding site on each was defined by one or more template ligands as indicated in each results section. The maximum number of poses was set to 80, retaining the highest scoring poses across all proteins. The ‘Normal’ docking mode used is optimized provide and accurate and exhaustive search. The default parameters of the LeadFinder™ genetic algorithm (pool size, population size and maximum constrain penalty) were used.

### 3.2. Solvation Thermodynamics and Gibbs Free Energies Calculations

The location and thermodynamic stability (ΔG values) of water molecules in the protein was investigated using the Reference Interaction Site Model (3D-RISM). Conceptually, this analytical method is equivalent to running an infinite time Molecular Dynamics simulation on the solvent with a fixed solute and then extracting the density of solvent particles, based on the Molecular Ornstein Zernike equation: *h*(*r*_12_) = *c*(*r*_12_) + ∫*dr*_3_ *c*(*r*_13_) *ρ*(*r*_3_)*h*(*r*_23_), where: *h*(*r*_12_) is the total correlation function (‘What is the distribution of the solvent around the solute’), *c*(*r*_12_) is the direct correlation function (‘How does a solvent molecule interact with the solute?’), and *dr*_3_ *c*(*r*_13_)*ρ*(*r*_3_)*h*(*r*_23_) is the indirect influence through all possible chains of mediating third particles (‘What is the effect of a solvent molecule interacting with another solvent molecule which is interacting with the solute?’).

Gibbs free energies (ΔG) were calculated in Flare 4.0.2. This function performs accurate estimation of the free energy of protein-ligand binding for a given complex. Besides the typical enthalpic energy terms, binding free energies include entropic contributions: A polar component of ligand desolvation upon binding using an adapted version of the Born model, energy penalties accounting for the accessibility of each H-bond donor/acceptor for water molecules and the strength of lost H-bonds upon ligand transfer from water to protein environment, also accounting for the loss of protein H-bonds induced by ligand binding, nonpolar solvation favoured by hydrophobic contacts in the complex, internal energy losses of the ligand upon transition from solvent to protein bound state, and entropic losses due to freezing ligand’s degrees of freedom upon binding. This method has demonstrated a rmsd of 1.5 kcal/mol with respect to observed binding constants [31].

### 3.3. Monte Carlo (MC) Energy Perturbations and Electrostatic Complementarity (EC)

Conformational search simulations by random perturbation of the torsional angles were carried out in AMMP 2.4.1(c) in VEGA ZZ 3.0.5 [32] using the MC Boltzmann jump method at a temperature of 1000 K, with a torsion rmsd of 60° to generate significantly different conformations at each step, followed by 20 energy minimization steps. This method allows upward jumps in energy to explore the conformational landscape and employs the Metropolis criterion to accept or reject perturbed conformations.

The *EC* was calculated in Flare 4.0.2 from the comparison of the protein and ligand electrostatic potential (*ESP*) based on the polarizable XED force field, describing atomic charge anisotropy. *ESP* values are generated and mapped at all vertices of a ligand or protein solvent accessible surface (SAS). Local visualization of the EC is allowed by the calculation of the *EC* score as follows: *EC* = ∫∫*_S_* [1 − *ESP_L_* + *ESP_P_*/*max* (*ESP_L_*, *ESP_P_*, *k*)] *dS*, where the integral is over the ligand *SAS*, *ESPL*, and *ESPP* are the ligand and protein *ESP* values, and max (*ESPL*, *ESPP*, *k*) is the protein or ligand *ESP* value with the largest deviation from zero, or a constant *k* if that is larger. A *k* value of 5 was chosen heuristically. Both *ESPL* and *ESPP* were capped to a maximum deviation from zero of 12 for the *EC* score. The capping value was derived from the XED *ESP*.

### 3.4. Ensemble Binding Space Analysis

Derived from property space analysis, the ensemble binding space analysis, developed by Vistoli et al. (2017), constitutes an approach to account for the dynamic processes of protein flexibility and ligand mobility. Briefly, this incorporates the statistically confirmed idea that alternative binding modes and the degree and ease of mobility of a ligand within a binding site significantly contribute to the observed affinity [24]. Accordingly, multiple low-energy docking poses are generated and analysed to calculate the relative frequencies of ligand interaction. The docking grid included all residues within 6 Å of the ligands unless otherwise specified. All elucidated structures of the hSERT from the PDB database at RCSB were included for ensemble binding space analysis.

## 4. Conclusions

MDMA occupation of the allosteric site (S2) of the hSERT is energetically favourable and it triggers protein readjustments of known allosteric regulation under Monte Carlo (MC) simulations. Upon association to S2, MDMA and a group of hSERT releasers can adopt two orientations. One that coincides with the binding site of 5-HT and the other with that of escitalopram. The concomitant occupation of these sites by MDMA may be possible and intermediate conformations were identified. The experimental activities of a set of psychoactive hSERT inhibitors were quantitatively correlated to their computed ΔG values for the orthosteric (central) and the allosteric binding solutions, thereby validating the modelling in principle. Based on ensemble binding analyses on all reported hSERT structures in distinct conformations and ΔG calculations, we proposed a pathway for MDMA from the extracellular vestibule to the orthosteric site. The main residues identified in the allosteric binding of MDMA lie between transmembrane segments TM10, TM11, TM6, and TM1b, i.e., Glu494, Asp328, Phe556 Phe335, Arg104, Thr497, and Gln332. These molecular models may be used for virtual screening and may pave the way to the identification and development of allosteric modulators of the hSERT with a MDMA-like chemotype. In conjunction with previous computational findings by our group [Islas et al. (2021)], these results constitute a structure-based mechanistic hypothesis for some of the molecular determinants that underlie the action of MDMA on the hSERT.

## Figures and Tables

**Figure 1 molecules-27-02977-f001:**
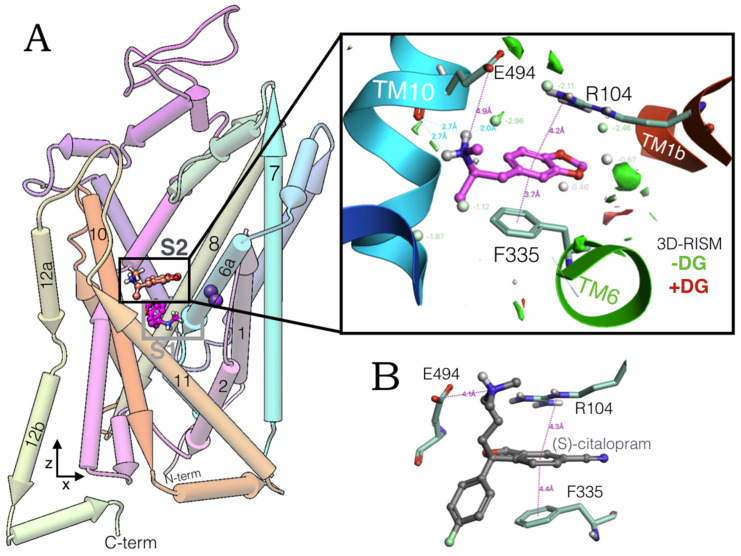
(**A**) The human serotonin transporter (hSERT) bound to two molecules of MDMA at the central (S1) and the allosteric binding site (S2), midway between the intracellular and extracellular side and on the extracellular vestibule, respectively. Cylinders represent the transmembrane segments. The insert shows the binding residues associated to their bond distance with 3D-RISM solvation in a green (energetically favourable) to red (energetically disfavourable) colour scale. (**B**) Binding mode of escitalopram from the intact complex PDB:5I73.

**Figure 2 molecules-27-02977-f002:**
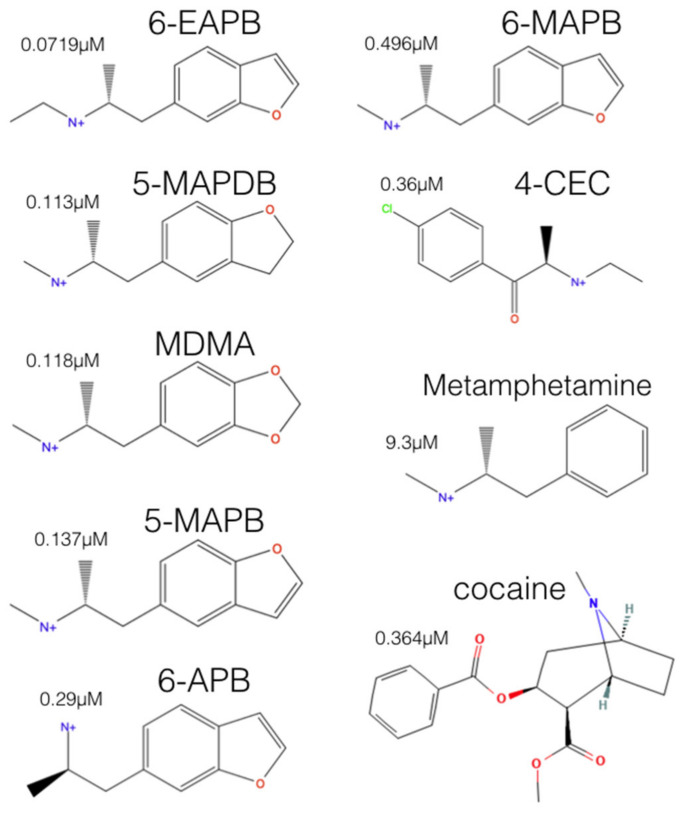
Selected hSERT blockers to simulate the ligand-protein interaction of S1 and S2 of the hSERT associated to their [^3^H]5-HT uptake inhibitory potencies (IC_50_s) from functional reuptake assays of HEK-293 cells stably expressing the hSERT. Experimental values taken from [7].

**Figure 3 molecules-27-02977-f003:**
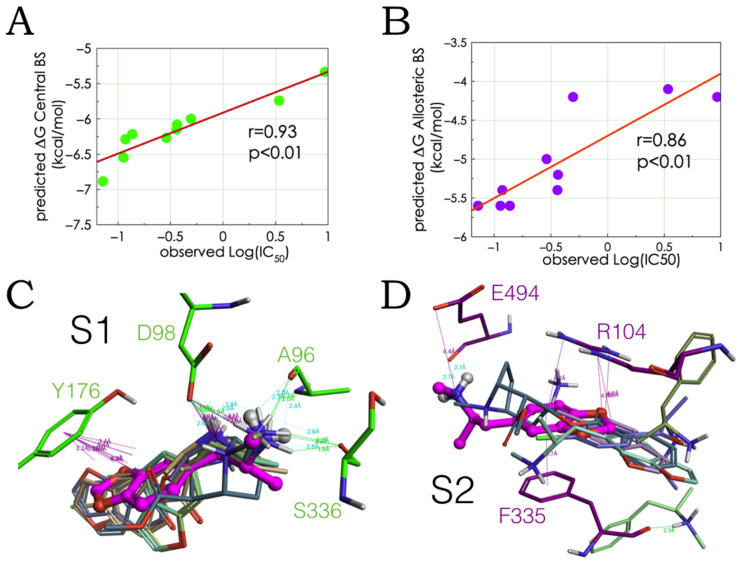
Correlation between experimental and computed data. The calculated Gibbs free energies of drugs in Figure 3, bound to S1 (**A**) or S2 (**B**) are plotted as a function of their [^3^H]5-HT reuptake block activities. The linear regressions (y = x × a + b) are associated to Pearson correlation coefficients. Binding modes of these compounds to S1 (**C**) and to S2 (**D**) of the hSERT. Ionic and aromatic bonds (in purple) and H-bonds (in cyan = weak, green = strong) are associated to their distances. MDMA is in ball and stick representation in magenta.

**Figure 4 molecules-27-02977-f004:**
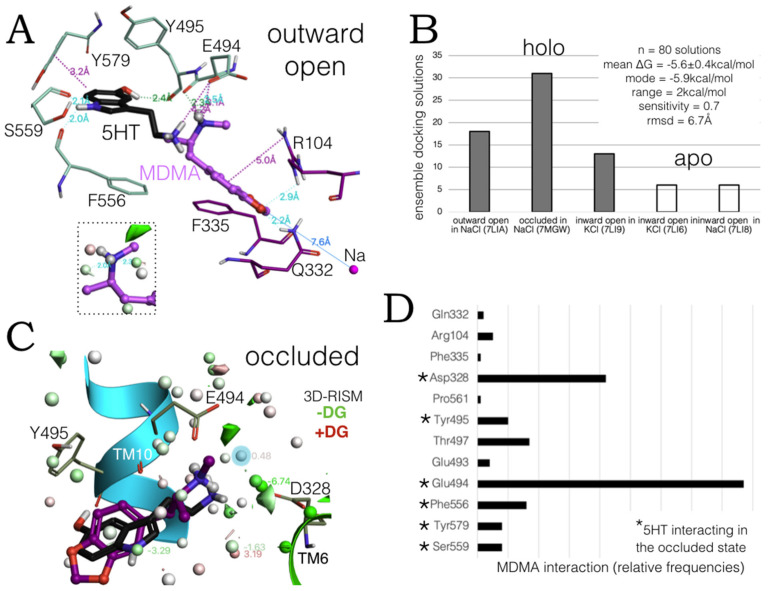
Ensemble binding space analysis of MDMA at the allosteric site of the hSERT. (**A**) Induced fit model of MDMA binding to the holo outward open state obtained in complex with 5-HT (in black). The sodium ion located between TM segments 1a and 1b is shown. Dotted lines represent intermolecular bonds associated to their distances. Ionic and aromatic bonds (in purple) and H-bonds (in cyan = weak, green = strong) are associated to their distances. The 3D-RISM solvation of the methylammonium group of MDMA is shown in the insert. (**B**) Number of energetically favourable MDMA poses per hSERT structure on S2. PDB codes are given in parenthesis. (**C**) The most representative binding pose from the cluster analysis of MDMA (in purple) on the occluded state of the hSERT with 3D-RISM solvation, waters are associated to their ΔG values. The intact position of 5-HT at S2 is in black. (**D**) Relative occurrences of the repertoire of MDMA interacting residues at the allosteric site. Asterisks denote the residues that interact with 5-HT at S2 on the occluded conformation.

**Figure 5 molecules-27-02977-f005:**
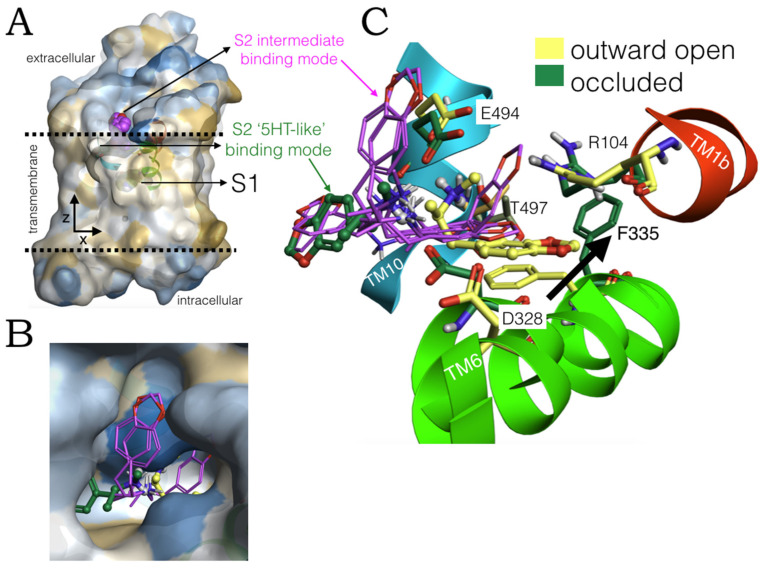
(**A**) Molecular surface of the hSERT complex in the occluded state showing the locations of MDMA at the central site and at S2. Including an intermediate configuration between the ‘5-HT-like’ and the ‘escitalopram-like’ orientation (in magenta). (**B**) Molecular surface of the binding pocket of S1 in the occluded state. The ‘5-HT-like’ binding mode in green, the ‘escitalopram-like’ binding mode in yellow and intermediate modes in magenta. Colour code: beige = hydrophobic, blue = polar. (**C**) Binding poses on S2 in the open outward and occluded states, ‘5-HT-like’ binding mode is in green, ‘escitalopram-like’ mode is in yellow, intermediate modes in magenta. The main participating binding residues are shown in the same colour code.

**Figure 6 molecules-27-02977-f006:**
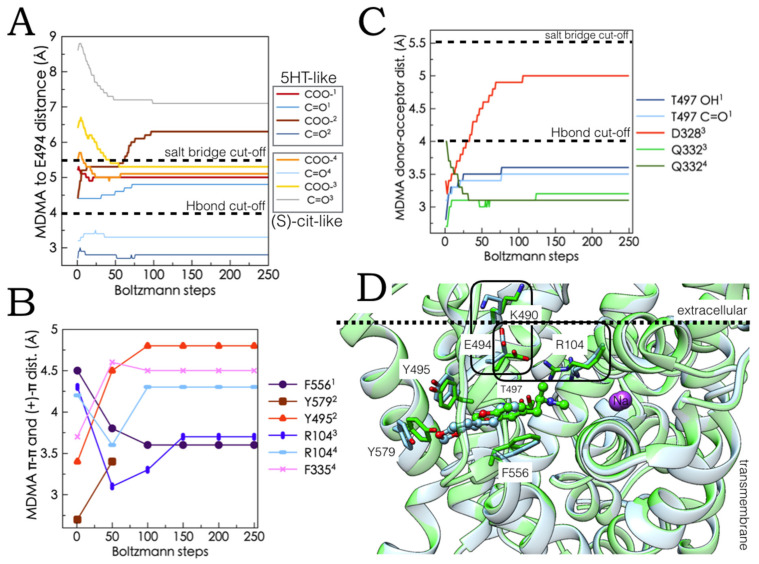
Ligand-protein electrostatic interactions under Monte Carlo (MC) simulations of four holo double-bound MDMA/hSERT complexes. Data correspond to either the ‘5-HT-like’ (replicas 1 and 2 in superscript) or the ‘escitalopram-like’ orientation (replicas 3 and 4 in superscript). (**A**) Interatomic distances from the cationic amine of MDMA and either the oxygen from the side chain (carboxylate) or the backbone (carbonyl) of Glu494 along the MC trajectories, as a marker of stability and occurrence of electrostatic bonds (according to the rectangles in the right-hand side of the plot). (**B**) Occurrence and stability of aromatic interactions evaluated by centroid-to-centroid distances with a 5 Å cut-off, as a function of the MC energy jumps. Data points represent allowed π interactions. (**C**) Interatomic ligand-protein H-bond/salt bridge distances as a function of the MC energy jumps (*n* = 4 in a total of 1000 Boltzmann steps). (**D**) MDMA/hSERT complexes in the ‘5-HT-like’ orientation of the drug at the end of the MC simulations. Replica 1 in blue and replica 2 in green. MDMA corresponding molecules are in ball and stick representation. The residues with the most favourable interactions in this orientation are shown. The salt bridges involved in the allosteric modulation of the hSERT are black rectangles.

**Figure 7 molecules-27-02977-f007:**
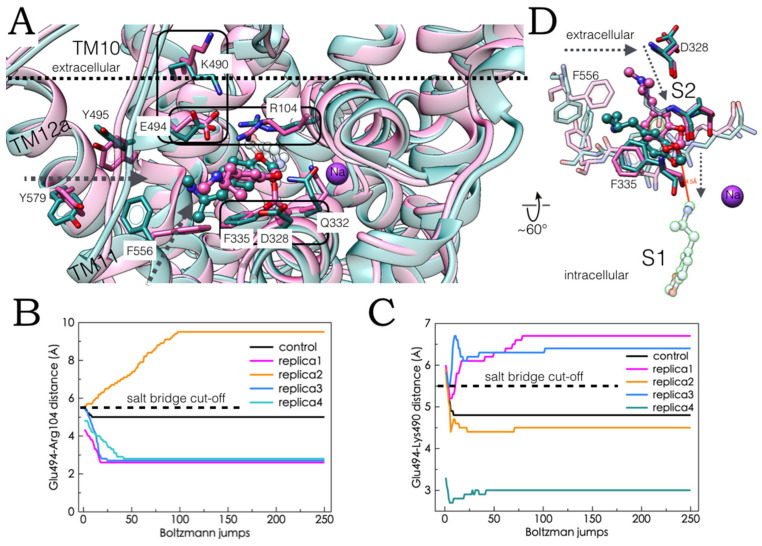
(**A**) MDMA/hSERT double-bound complexes subjected to Monte Carlo (MC) simulations in the ‘escitalopram-like’ orientation of the drug. Replica 3 in pink, replica 4 in blue. MDMA corresponding molecules are in ball and stick representation, MDMA at S1 is in translucent blue. The residues proposed to participate in the allosteric binding of MDMA are shown. The residues involved in the allosteric modulation of the hSERT are black rectangles. Donor-to-acceptor distance as a function of the MC energy perturbation steps for the allosterically coupled salt bridges between (**B**) Glu494 and Arg104, and (**C**) Glu494 and Lys490. (**D**) Rotated view of (**A**), in the absence of TM segments (ribbons) showing the proposed entry pathway for MDMA from S2 to S1.

**Figure 8 molecules-27-02977-f008:**
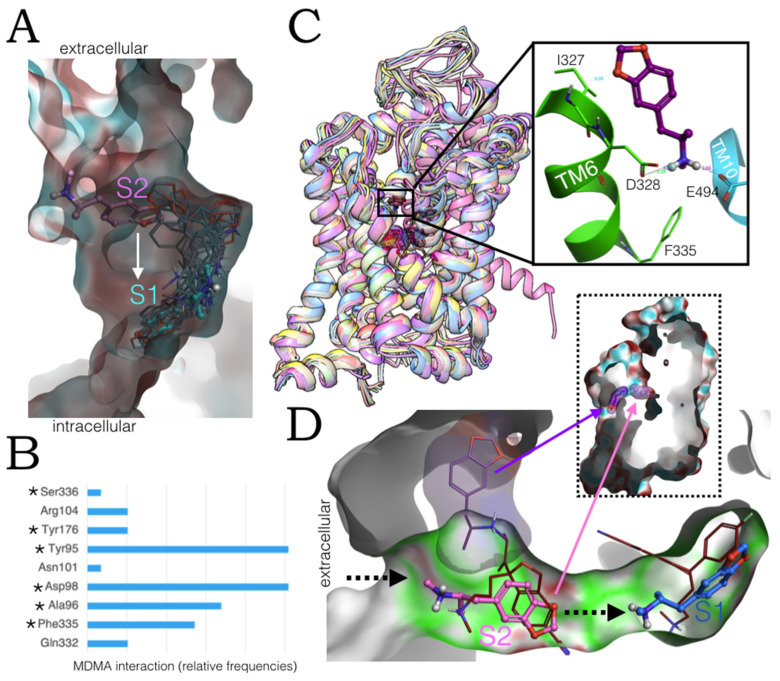
Ensemble binding space of MDMA on the allosteric and central site of hSERT. (**A**) Electrostatic surface of the MC hSERT complex model from the external vestibule S2 (in pink) to the central site (in cyan) intermediate conformations in thin sticks in grey. Surface colour scale: light blue = negative, red = positive charge. (**B**) Bonding incidences of the intermediate conformations of MDMA found on the double-bound complex after MC simulations. Asterisks mark the residues involved in the binding of MDMA to the central site. (**C**) Top 20 most energetically favoured poses from ensemble docking to all available hSERT structures. The insert shows a ‘flipped’ intermediate conformation on the extracellular vestibule of S2. (**D**) Electrostatic complementarity surface of a MC double-bound hSERT/MDMA model showing an alternative location of MDMA at S2 with a ‘5-HT-like orientation’. Intact escitalopram locations are shown in thinner dark red sticks.

**Figure 9 molecules-27-02977-f009:**
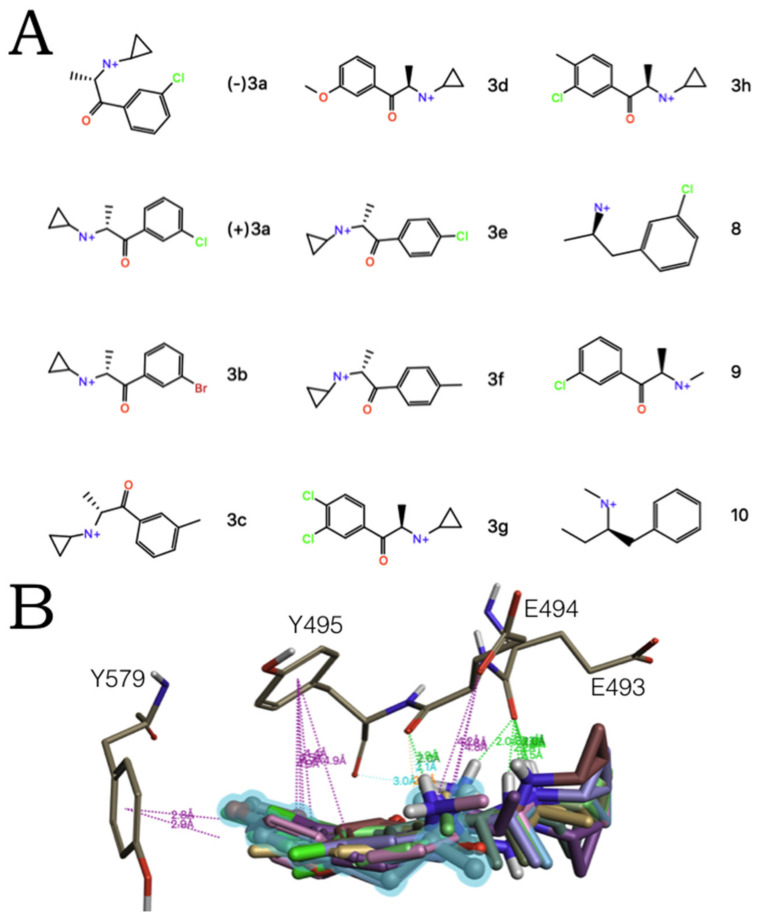
(**A**) 5-HT releasing compounds selected to test the MDMA/hSERT allosteric model. (**B**) Binding of these molecules to the allosteric site of the hSERT model in the ‘5-HT-like orientation’ of MDMA (highlighted in blue). Dotted lines represent intermolecular bonds associated to their distances. H-bonds are in cyan and a stronger H-bond in green. Salt bridges and π-π interactions are in purple.

**Table 1 molecules-27-02977-t001:** Gibbs free energies of compounds in Figure 9A on the allosteric site of the MDMA/hSERT models.

5-HT-Like Orientation	Escitalopram Orientation	*p*-Value
−7.1 ± 0.9 kcal/mol *	−6.3 ± 0.6 kcal/mol	0.03

Means ± SD. Significance obtained with a *t*-test. * *p* < 0.05.

## Data Availability

Key model structures (.pdb files) are available (free of charge for non-profit use) upon request.

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
