# Peer review of "Allosteric Binding of MDMA to the Human Serotonin Transporter (hSERT) via Ensemble Binding Space Analysis with ΔG Calculations, Induced Fit Docking and Monte Carlo Simulations"

_molecules, 2022, doi:10.3390/molecules27092977_

Round 1
Reviewer 1 Report
The authors used different theoretical methods with intention to study different allosteric hSERT ligands. Overall, the manuscript is well written. However, I have some minor revisions:
1) In the Abstract, the authors should say the meaning of the MDMA.
2) The titles of the sections are long, the author should correct them.
3) In all figures, the authors discuss the results and these must be inserted in the text and not in the figures.
4) What the distance used to to describe the interatomic donor-acceptor, pi-pi or cation-pi? Only this criteria is sufficient to do this? For instance, these interactions must be confirmed by quantum chemistry calculations.
5) The references of the Flare4.0 and VEGA ZZ 3.0.5 programs are missing.
6) The reference of Islas et al., (2021) is missing.
7) The references in the text are in a different format from the Reference section.
Author Response
Response to Reviewer 1 Comments
Point 1: In the Abstract, the authors should say the meaning of the MDMA.
Response1: this has been corrected
Point 2: The titles of the sections are long, the author should correct them.
Response 2: We have edited the titles and they are now shorter
Point 3: In all figures, the authors discuss the results and these must be inserted in the text and not in the figures.
Response 3: efforts have been made to reduce the length of the captions, leaving only essential descriptions for every figure and all discussion in the text
Point 4: What the distance used to to describe the interatomic donor-acceptor, pi-pi or cation-pi? Only this criteria is sufficient to do this? For instance, these interactions must be confirmed by quantum chemistry calculations.
Response 4: distance constrains for donor-acceptor H-bond, slat bridge, pi-pi and cation-pi are according to proprietary software Flare4.0 (Cresset®, Litlington, Cambridgeshire, UK). Citation: Cheeseright T, Mackey M, Rose S, Vinter A. Molecular field extrema as descriptors of biological activity: definition and validation. J Chem Inf Model. 2006 Mar-Apr;46(2):665-76.
Point 5: The references of the Flare4.0 and VEGA ZZ 3.0.5 programs are missing.
Response 5: these references have been added
Point 6: The reference of Islas et al., (2021) is missing.
Response 6: this reference has been added
Point 7: The references in the text are in a different format from the Reference section.
Response 7: this has been corrected
Reviewer 2 Report
The reference [Islas et al., (2021)] from text is not found in the bibliography at the end of the manuscript
Author Response
Response to Reviewer 2 Comment
Point 1: The reference [Islas et al., (2021)] from text is not found in the bibliography at the end of the manuscript
Response 1: this reference has been added
Reviewer 3 Report
The description of the MDMA binding to the hSERT presented by Islas and Scior is very detailed and can definitely be used by other researchers for comparison and drug development. However there are several technical and scientific issues that should be addressed before the publication:
1) At fig. 3A-3B authors refer to "correlation", however they present a linear regression plot. First, if it is a true correlation, then a proper statistical metric such as Pearson or Spearman correlation coefficient should be used. Moreover, from the looks of the graph, the linear model is not the best choice here. Maybe log-transformation of the Y-axis (or both axes) would bring the data to a more linear shape. But in the current state, the conclusion "Model results were in good keeping with experimental data (R2 =0.65)." is a huge overstretch.
2) Since authors use proprietary software, the description of methods has to be improved. Especially the docking section: details like the treatment of heteroatoms and water molecules, altlocs (if applicable), protonation states of both protein and ligands, etc. All those details would be needed by anyone, who would want to reproduce their work using other tools widely used by the community.
3) Because of the use of the proprietary software, it would be very hard to reproduce the results, thus authors should also release key structures (including those, referred to as intermediate binding structures) as files in commonly used formats such as PDB or mmCIF. This would allow other researchers to compare future results, molecules, and binding poses with the ones, obtained by authors.
Author Response
Response to Reviewer 3
Point 1: At fig. 3A-3B authors refer to "correlation", however they present a linear regression plot. First, if it is a true correlation, then a proper statistical metric such as Pearson or Spearman correlation coefficient should be used. Moreover, from the looks of the graph, the linear model is not the best choice here. Maybe log-transformation of the Y-axis (or both axes) would bring the data to a more linear shape. But in the current state, the conclusion "Model results were in good keeping with experimental data (R2 =0.65)." is a huge overstretch.
Response 1: We agree and thank the reviewer for this valuable observation. The plots have been modified with the log-transformation and the Pearson correlation coefficients are reported (r=0.93 and 0.86 for the central and the allosteric binding sites, respectively).
Point 2: Since authors use proprietary software, the description of methods has to be improved. Especially the docking section: details like the treatment of heteroatoms and water molecules, altlocs (if applicable), protonation states of both protein and ligands, etc. All those details would be needed by anyone, who would want to reproduce their work using other tools widely used by the community.
Response 2: Those and more details were added in section 3.1
Point 3: Because of the use of the proprietary software, it would be very hard to reproduce the results, thus authors should also release key structures (including those, referred to as intermediate binding structures) as files in commonly used formats such as PDB or mmCIF. This would allow other researchers to compare future results, molecules, and binding poses with the ones, obtained by authors.
Response 3: We understand the concern of the reviewer and a data availability statement has been added at the end of the manuscript.